# Seroepidemiologic Survey of Lyme Disease among Forestry Workers in National Park Offices in South Korea

**DOI:** 10.3390/ijerph18062933

**Published:** 2021-03-12

**Authors:** Dilaram Acharya, Ji-Hyuk Park

**Affiliations:** Department of Preventive Medicine, College of Medicine, Dongguk University, Gyeongju 38066, Korea; dilaramacharya123@gmail.com

**Keywords:** Lyme disease, seroprevalence, risk factors, forestry workers

## Abstract

Limited data are available on the current status of Lyme disease in South Korea. The aim of this study was to investigate the seroprevalence and risk factors associated with Lyme disease infection among forestry workers in National Park Offices in South Korea. We enrolled National Park Office forestry workers (NPOFWs) who had worked for ≥1 year. Participants completed questionnaires that addressed various subjects including work types and work hygiene-related factors. Collected serum samples were tested using immunofluorescence assay to detect anti-*Borrelia* antibodies. Multivariate logistic regression was used to identify independent risk factors of seroprevalence. Of 1,410 NPOFWs, 655 (46.5%) participated in this study, and an overall seroprevalence of Lyme disease antibodies was 8.1%. Analysis showed that always eating meals in woodland (odds ratio (OR), 5.11; 95% confidence interval (CI), 2.08–12.52) and raising dogs outside homes (OR, 3.25; 95% CI, 1.57–6.75) were significantly associated with Lyme disease infection. This seroprevalence study indicates that Lyme disease is an important disease among NPOFWs in South Korea. These identified modifiable risk factors should be considered while designing preventive strategies for reducing Lyme disease infection among NPOFWs.

## 1. Introduction

Lyme disease is a zoonotic tick-borne disease that is caused by a wide variety of spirochete *Borrelia burgdorferi*. In the Americas, *B. burgdorferi* sensu stricto (s.s.) are the main pathogen, while in Europe, *B. afzelii* and *B. garinii* are common [1]. Lyme disease is a multi-system illness that primarily affects the nervous system, skin, joints, and heart. Most people treated with antibiotics for Lyme disease do recover whereas only a small percentage have persisting symptoms [2]. Delayed treatment can result in sequelae and long-term antibiotic use, and results in excessive health care burdens in terms of cost and the use of health facilities [3].

The incidence of Lyme disease has been consistently reported to be increasing in different parts of the world including America, Europe, and Asia [4]. The risk of contracting the disease is largely dependent on the activities of individuals and exposure in areas inhabited by ticks, which include woodlands, rural areas, and forests [5,6,7,8]. Several studies reported that a wide range of personal and environmental preventive measures, such as wearing protective clothing, checking for ticks during outdoor activities, applying repellents prior to outdoor activities, taking a shower after visiting areas at risk, and the use of host-targeted acaricides, are protective factors for Lyme disease infection [5,7,9].

In South Korea, Lyme disease was designated as a National Notifiable Infectious Disease in 2010, and since then, reports of the disease have increased. During 2011–2019, an average of 15.8 cases of Lyme disease was reported annually, with 23 cases in 2019 [10]. A rise in cases of Lyme disease [10] and the identification of *B. burgdorferi* infection in animals [11,12] in South Korea suggest that Lyme disease will continue to be a public health problem. Employees assigned to work in forest, woodland, and suburban areas are more likely to be bitten by ticks, and thus, are at greater risk of Lyme disease [13,14]. Few human studies on Lyme disease have been published in South Korea [15,16], however, these studies did not investigate Lyme disease infection among forestry workers.

Investigations of the prevalence and risk factors for Lyme disease infection among risk groups aid the design and implementation of effective preventive strategies. The aim of this study was to determine the seroprevalence and identify the risk factors associated with Lyme disease infection among forestry workers in national parks, who constitute a high-risk group in South Korea.

## 2. Materials and Methods

### 2.1. Ethics and Consent

This study was conducted after obtaining approval from the Institutional Review Board of Dongguk University Gyeongju Hospital (approval number: 16-286). Written informed consent was obtained from all study participants prior to enrollment. All personal identifiers were removed before data analysis, and privacy, confidentiality, and anonymity were fully maintained.

### 2.2. Study Setting and Population

The Korea National Park Service was established in 1987 to conserve natural resources through research and study. This organization is responsible for the management of national parks, which involves, for example, the installation and maintenance of park facilities, the publication of maps, and promotional activities [17]. In South Korea, the National Park Service manages 21 of the 22 national parks, except the Hallasan National Park in Jeju Island. National Park Offices are affiliated with the Korea National Park Service and are located near national parks to facilitate professional management. In view of the low incidence of Lyme disease in South Korea (0.02 cases per 100,000 population in 2015) [10], we restricted participants to those who had worked for ≥1 year as National Park Office forestry workers (NPOFWs) and enrolled 655 of 1410 NPOFWs working in 29 main offices and 65 branch offices throughout the country.

### 2.3. Data Collection

In cooperation with the Korea National Park Service, we informed NPOFWs about the study and mailed questionnaires to them prior to visiting offices. The self-developed questionnaire consisted of four parts: (i) general characteristics of participants; (ii) type of work performed; (iii) work hygiene-related factors (including wearing personal protective equipment, risky behaviors during outdoor work, and protective behaviors during and after outdoor work); and (iv) other potential risk factors (such as performing additional jobs and raising animals). Our study team visited all 29 main offices during December 2016; the enrolled workers were asked to visit the nearest main office for appointments. Completed questionnaires were verified by the researchers to ensure the quality of the data obtained.

### 2.4. Serologic Testing

On interview days, a blood sample (10 mL) was collected from each participant. Serum samples were transferred to the Korea Centers for Disease Control and Prevention to test for Lyme disease. An in-house indirect immunofluorescence assay (IFA) was performed to quantify immunoglobulin G (IgG) and immunoglobulin M (IgM) antibody levels to *B. burgdorferi* [18]. Samples were considered seropositive when IgG or IgM antibody titers were ≥1:256 or ≥1:16, respectively, which are the criteria used during the first-step of laboratory testing for Lyme disease in South Korea [18].

### 2.5. Statistical Analysis

Data were analyzed using the Statistical Package for the Social Sciences (SPSS) version 20.0 (SPSS, IBM, Armonk, NY, USA). We conducted univariate logistic regression analysis to identify factors associated with Lyme disease infection among NPOFWs. Following this, factors with *p*-value < 0.10 were incorporated into multivariate logistic regression with backward elimination to calculate odds ratios (ORs) and 95% confidence intervals (CIs). The statistical significance was set to *p*-value < 0.05.

## 3. Results

### 3.1. Personal Profiles of the Participants

The 655 NPOFWs comprised 502 (76.6%) men and 153 (23.4%) women. Median age was 43 years (range, 18–71), and median duration of working for National Park Offices was 7 years (range, 1–38). Four hundred and seventeen workers (63.7%) were based at main offices and 238 (36.3%) at branch offices.

### 3.2. Serologic Results

Six hundred and fifty-five (46.4%) of 1410 NPOFWs participated in this study. The overall seroprevalence of Lyme disease among participants was 8.1% (53/655). Titer endpoints for *B. burgdorferi* IgG varied between <1:16 and 1:512, and in 21 samples (3.2%) reached or went beyond 1:256. Titer endpoints for *B. burgdorferi* IgM varied between <1:16 and 1:32, and in 34 samples (5.2%) reached or went beyond 1:16. Two samples (0.3%) had IgG titers of ≥1:256 and IgM titers of ≥1:16 (Table 1).

### 3.3. Univariate Analysis of Lyme Disease Infection and Associated Risk Factors

Older individuals (≥50 years) were less likely to be seropositive to *B. burgdorferi* than younger individuals (<29 years) (*p* = 0.081). However, duration of work, sex, region, type of organization, and level of education were not associated with Lyme disease infection (Table 2). Although NPOFWs perform various tasks, work type was not associated with Lyme disease infection (Table 3). NPOFWs who always ate meals in woodland during outdoor work were associated with a higher risk of Lyme disease infection (*p* < 0.001). Additionally, NPOFWs who always showered after outdoor work had a lower risk of Lyme disease infection (*p* = 0.072, Table 4). Furthermore, workers who raised dogs outside homes were associated with a higher risk of Lyme disease infection (*p* = 0.005). However, raising dogs inside homes and cats were not associated with Lyme disease infection (Table 5).

### 3.4. Multivariate Analysis of Lyme Disease Infection and Associated Risk Factors

Potential factors associated with Lyme disease infection as determined by multivariate logistic regression analysis are shown in Table 6. Important variables (*p* < 0.10) identified by univariate analysis, that is, age, eating meals in woodland, taking a shower after outdoor work, and raising dogs outside homes, were incorporated into multivariate logistic regression analysis with backward elimination. Multivariate logistic regression analysis showed that always eating meals in woodland (OR, 5.11; 95% CI, 2.08–12.52) was significantly associated with a higher prevalence of Lyme disease infection. Furthermore, the risk of Lyme disease infection was significantly higher among NPOFWs who raised dogs outside homes (OR, 3.25; 95% CI, 1.57–6.75). However, taking a shower after outdoor work did not significantly reduce the risk of Lyme disease infection (OR, 0.60; 95% CI, 0.34–1.07).

## 4. Discussion

This is the first study conducted to assess the status of Lyme disease infection among forestry workers using nationally representative data in South Korea. In the present analysis, we have identified risk factors associated with Lyme disease infection among NPOFWs, and completed a preliminarily published information on this study [19]. An overall seroprevalence of Lyme disease was 8.1% as determined by IFA (IgG titer ≥1:256 or IgM titer ≥1:16). The seroprevalence, based on an IFA IgG titer ≥1:256 (3.2%), was lower than those reported in Europe among forestry workers, such as in France (15.2%) [20] and Slovenia (9.8%) [21]. A seroepidemiologic study performed in China among people living in forested areas reported Lyme disease seroprevalence ranging from 2.9% to 14.9% as determined by an IFA IgG titer of ≥1:128 [22], which is similar to the seroprevalence of 6.1% (IFA IgG titer ≥1:128) found in this study.

Participants who always ate meals in woodland during outdoor work had a higher odds ratio for seropositivity. Woodland areas with long grass and scrubs are preferred tick habitats, and ticks tend to be plentiful in these areas [23]. A Belgian study among forestry workers reported that the number of tick bites and the use of personal protective measures impacted the seroprevalence of Lyme disease [8]. However, eating meals in woodland was not included in that Belgian study, and we are unaware of any published paper reporting that eating meals in woodland is a possible risk factor of Lyme disease infection.

NPOFWs who raised dogs outside homes were found to be at higher risk of Lyme disease infection, whereas raising dogs inside homes was not associated with the presence of infection. In a previous study on tick exposure risk factors, the presence of indoor/outdoor pets was positively associated with the presence of ticks [14]. Another Chinese study reported that pets in households were not significantly associated with the seroprevalence of Lyme disease [24]. Thus, it appears that dogs kept outside homes can harbor ticks infected with *B. burgdorferi* and transfer these ticks to their owners. Lyme disease seroprevalence among dogs and cats may be a sentinel marker of human Lyme disease infection because they share the environment with their owners [25,26]. Furthermore, *B. burgdorferi* infection in dogs has been consistently reported in many countries including South Korea [11,25]. Therefore, we presume that our study participants who were engaged in eating meals in woodland and raising dogs outside homes might have been exposed to tick bites that resulted in the seropositive test results.

Regarding work hygiene-related factors, we found that a higher percentage of participants that always showered after outdoor work were seronegative, though this association did not reach statistical significance in the multivariate analysis. Several previous studies [5,7] have reported reduced risks of Lyme disease among those who use protective measures, such as protective clothing or repellents, and among those that check for ticks during outdoor activities or shower regularly after outdoor work, which would presumably remove any ticks [14]. In this study, the proportions of NPOFWs always using insect repellents (7.4%), using a mat to rest (8.5%), and checking tick bites after outdoor work (15.4%) were relatively low, and the proportion of recognition of Lyme disease was also low (16.8%). Thus, health education about Lyme disease and personal protection to prevent tick bites needs to be strengthened.

The present study has several limitations that warrant consideration. First, seroprevalence may have been influenced by seasonality since our study was performed in the winter (December) to maximize NPOFW participation, when ticks are least active. Second, NPOFWs on Jeju Island (South Korea) were not included. Third, we could not investigate the Lyme disease infection statuses of dogs raised outside homes. Fourth, seropositive test results might not necessarily be equivalent to infection, but rather may simply be evidence of past exposure. However, this is one of the important studies being conducted with a large sample and our findings can have policy implications in South Korea and other similar settings elsewhere.

## 5. Conclusions

The overall seroprevalence of Lyme disease among NPOFWs in South Korea was found to be 8.1%. Eating meals in woodland and raising dogs outside homes were associated with higher risks of Lyme disease infection. These modifiable risk factors should be considered when designing preventive strategies aimed at reducing Lyme disease infection. Further comparative seroepidemiological and longitudinal studies are required among different populations at risk of Lyme disease infection.

## Figures and Tables

**Table 1 ijerph-18-02933-t001:** Serologic results for *Borrelia burgdorferi* antigen among National Park Office forestry workers in South Korea.

Titer	IgG		IgM	
	No.	%	No.	%
<1:16	196	29.9	621	94.8
1:16	215	32.8	25	3.8
1:32	136	20.8	9	1.4
1:64	68	10.4	0	0.0
1:128	19	2.9	0	0.0
≥1:256	21	3.2	0	0.0
Total	655	100.0	655	100.0

**Table 2 ijerph-18-02933-t002:** Univariate logistic regression analysis between demographic characteristics and Lyme disease seroprevalence among National Park Office forestry workers in South Korea.

Variables	Total	Seroprevalence No. (%)	OR (95% CI)	*p*-Value
Sex				
Men	502	39 (7.8)	0.84 (0.44–1.59)	0.584
Women	153	14 (9.2)	Reference	
Age (years)				
<29	76	8 (10.5)	Reference	
30–49	350	34 (9.7)	0.91 (0.41–2.06)	0.830
≥50	229	11 (4.8)	0.43 (0.17–1.11)	0.081
Duration of work (years)				
<5	232	16 (6.9)	Reference	
5–<15	288	28 (9.7)	1.45 (0.77–2.76)	0.252
≥15	135	9 (6.7)	0.96 (0.41–2.25)	0.933
Region				
Northern	134	9 (6.7)	Reference	
Central	260	17 (6.5)	0.97 (0.42–2.24)	0.946
Southern	261	27 (10.3)	1.60 (0.73–3.51)	0.239
Organization types				
Main offices	417	33 (7.9)	Reference	
Branch offices	238	20 (8.4)	1.07 (0.60–1.91)	0.825
Education				
High school or less	225	18 (8.0)	0.98 (0.54–1.77)	0.944
University or more	429	35 (8.2)	Reference	

OR, odds ratio; CI, confidence interval.

**Table 3 ijerph-18-02933-t003:** Univariate logistic regression analysis between work type and Lyme disease seroprevalence among National Park Office forestry workers in South Korea.

Variables	Total	Seroprevalence No. (%)	OR (95% CI)	*p*-Value
Monitoring of natural resources				
Yes	270	20 (7.4)	0.85 (0.48–1.52)	0.584
No	384	33 (8.6)	Reference	
Repairing facilities				
Yes	431	38 (8.8)	1.34 (0.72–2.49)	0.355
No	223	15 (6.7)	Reference	
Supervision of illegal activities				
Yes	392	33 (8.4)	1.11 (0.62–1.98)	0.719
No	262	20 (7.6)	Reference	
Patrolling				
Yes	543	45 (8.3)	1.16 (0.53–2.54)	0.704
No	111	8 (7.2)	Reference	
Guiding visitors				
Yes	407	35 (8.6)	1.20 (0.67–2.17)	0.542
No	248	18 (7.3)	Reference	
Exploration program				
Yes	152	16 (10.5)	1.48 (0.80–2.75)	0.211
No	503	37 (7.4)	Reference	
Grass mowing				
Yes	295	25 (8.5)	1.09 (0.62–1.92)	0.753
No	359	28 (7.8)	Reference	
Cleaning				
Yes	334	26 (7.8)	0.92 (0.52–1.61)	0.760
No	320	27 (8.4)	Reference	

OR, odds ratio; CI, confidence interval.

**Table 4 ijerph-18-02933-t004:** Univariate logistic regression analysis between work hygiene-related factors and Lyme disease seroprevalence among National Park Office forestry workers in South Korea.

Variables	Total	Seroprevalence No. (%)	OR (95% CI)	*p*-Value
During outdoor work				
Wearing a long-sleeved shirt				
Always	210	17 (8.1)	1.03 (0.56–1.88)	0.925
Others	444	35 (7.9)	Reference	
Wearing long pants				
Always	448	33 (7.4)	0.74 (0.42–1.33)	0.318
Others	207	20 (9.7)	Reference	
Wearing gloves				
Always	297	28 (9.4)	1.38 (0.79–2.43)	0.260
Others	357	25 (7.0)	Reference	
Wearing boots				
Always	327	31 (9.5)	1.45 (0.82–2.57)	0.199
Others	327	22 (6.7)	Reference	
Wearing a hat				
Always	258	23 (8.9)	1.19 (0.68–2.11)	0.540
Others	396	30 (7.6)	Reference	
Using insect repellents				
Always	26	4 (15.4)	2.14 (0.71–6.46)	0.177
Others	626	49 (7.8)	Reference	
Resting on the grass				
Always	14	2 (14.3)	1.93 (0.42–8.85)	0.398
Others	641	51 (8.0)	Reference	
Using a mat to rest				
Always	62	4 (6.5)	0.76 (0.27–2.19)	0.617
Others	592	49 (8.3)	Reference	
Eating meals in woodland				
Always	29	8 (27.6)	4.91 (2.06–11.71)	<0.001
Others	625	45 (7.2)	Reference	
Defecating/urinating in woodland				
Always	8	2 (25.0)	3.90 (0.77–19.80)	0.101
Others	647	51 (7.9)	Reference	
After outdoor work				
Taking a shower				
Always	386	25 (6.5)	0.60 (0.34–1.05)	0.072
Others	269	28 (10.4)	Reference	
Taking a bath				
Always	168	14 (8.3)	1.04 (0.55–1.97)	0.899
Others	486	39 (8.0)	Reference	
Changing working clothes daily				
Always	231	18 (7.8)	0.94 (0.52–1.70)	0.836
Others	424	35 (8.3)	Reference	
Washing working clothes daily				
Always	229	16 (7.0)	0.79 (0.43–1.45)	0.448
Others	426	37 (8.7)	Reference	
Checking tick bites				
Always	102	9 (8.8)	1.12 (0.53–2.37)	0.772
Others	552	44 (8.0)	Reference	

OR, odds ratio; CI, confidence interval.

**Table 5 ijerph-18-02933-t005:** Univariate logistic regression analysis between other work-related factors and Lyme disease seroprevalence among National Park Office forestry workers in South Korea.

Variables	Total	Seroprevalence No. (%)	OR (95% CI)	*p*-Value
Additional jobs				
Rice farming				
Yes	21	3 (14.3)	1.95 (0.55–6.83)	0.299
No	634	50 (7.9)	Reference	
Dry field farming				
Yes	93	8 (8.6)	1.08 (0.49–2.37)	0.845
No	562	45 (8.0)	Reference	
Orchard farming				
Yes	26	3 (11.5)	1.51 (0.44–5.21)	0.514
No	629	50 (7.9)	Reference	
Livestock farming				
Yes	13	1 (7.7)	0.95 (0.12–7.42)	0.957
No	642	52 (8.1)	Reference	
Raising animals				
Dogs (outside homes)				
Yes	70	12 (17.1)	2.75 (1.37–5.52)	0.005
No	585	41 (7.0)	Reference	
Dogs (inside homes)				
Yes	52	5 (9.6)	1.23 (0.47–3.24)	0.675
No	603	48 (8.0)	Reference	
Cats				
Yes	28	2 (7.1)	0.87 (0.20–3.77)	0.851
No	627	51 (8.1)	Reference	
Recognition of Lyme disease				
Yes	110	10 (9.1)	1.17 (0.57–2.40)	0.674
No	545	43 (7.9)	Reference	
Recognition of tick bites				
Yes	75	8 (10.7)	1.42 (0.64–3.13)	0.390
No	579	45 (7.8)	Reference	

OR, odds ratio; CI, confidence interval.

**Table 6 ijerph-18-02933-t006:** Multivariate logistic regression analysis of important variables (*p* < 0.10) associated with Lyme disease seroprevalence among National Park Office forestry workers in South Korea.

Variables	OR (95% CI)	*p*-Value
Age (years)		
<29	Reference	
30–49	0.98 (0.42–2.30)	0.971
≥50	0.41 (0.15–1.10)	0.078
Eating meals in woodland during outdoor work		
Always	5.11 (2.08–12.52)	<0.001
Others	Reference	
Taking a shower after outdoor work		
Always	0.60 (0.34–1.07)	0.086
Others	Reference	
Raising dogs (outside homes)		
Yes	3.25 (1.57–6.75)	0.002
No	Reference	

OR, odds ratio; CI, confidence interval.

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
