# Peer review of "Seroepidemiologic Survey of Lyme Disease among Forestry Workers in National Park Offices in South Korea"

_ijerph, 2021, doi:10.3390/ijerph18062933_

Round 1
Reviewer 1 Report
I am happy with the edits the authors have made to the manuscript. I detected one error on line 274. Please correct the statement regarding risks in older forestry workers by changing "more" to "less" in reference to their risks of being seropositive. In Table 2, the OR for employers older than 50 is less than 1.0, meaning their risk is lower than the risk in the <29 group.
Author Response
Response to Reviewer 1 Comments
Thank you for reviewing this article and giving us careful comments. We have revised the manuscript based on the given comments. All changes have been indicated with red colored writings in the manuscript.
Point 1: I am happy with the edits the authors have made to the manuscript. I detected one error on line 274. Please correct the statement regarding risks in older forestry workers by changing "more" to "less" in reference to their risks of being seropositive. In Table 2, the OR for employers older than 50 is less than 1.0, meaning their risk is lower than the risk in the <29 group.
Response 1: We corrected the statement according to your comment (Line 116).
Reviewer 2 Report
Seroepidemiologic survey of Lyme disease among forestry workers in National Park Offices in South Korea
Dear Authors,
This is quite an interesting study but needs some improvement to better represent the results.
In research articles, it is a good idea to introduce the research group first to get to know the subject of the research. Then the overall results (univariate analysis, seroepidemiological analysis) should be presented in sequence, and finally the results of the multivariate analysis. This order of things makes it easier for the readers to understand what the article is about.
Thank you.
Author Response
Response to Reviewer 2 Comments
Thank you for reviewing this article and giving us careful comments. We have revised the manuscript based on the given comments. All changes have been indicated with red colored writings in the manuscript.
Point 1: This is quite an interesting study but needs some improvement to better represent the results. In research articles, it is a good idea to introduce the research group first to get to know the subject of the research. Then the overall results (univariate analysis, seroepidemiological analysis) should be presented in sequence, and finally the results of the multivariate analysis. This order of things makes it easier for the readers to understand what the article is about.
Response 1: We modified the order of results according to your comments (Line 101-114).
Reviewer 3 Report
Dilaram Acharya & Ji-Hyuk Park: Seroepidemiologic survey of Lyme disease among forestry workers in National Park Offices in South Korea v. #2
I acknowledge that the most critical issues have been addressed. There remain, however, some minor ones that still need corrections.
P.1. l. 47 vs. l.57 (and throughout the text): a small terminological inconsistency – “forestry-“ or “forest workers” ? Both terms are used indiscriminately in the text. Are those employee just forest craftsmen/lumberjacks or are they responsible for forest management/conservation in a broader sense? Perhaps a neutral term, e.g. “forest and conservation workers”, would fit…
P.1., l.60: ..tested using immunofluorescence assay..
P.1, l.72: Lyme disease is a zoonotic tick-borne disease..
P.1, l.72-3: ..caused by a wide variety of the spirochaete..
P.1, l.87: comma after “acaricides”
P.2, l.163: ..obtaining an approval..
P.2, l.165: ..consents were obtained..
P.2, l.188: ..the enrolled workers were asked..
P.2, l.189: ..verified by the researchers..;
P.2, l.192: delete “also”, pls
P.2, l.196: “B. burgdorferi s.s.” (??) – an inhouse strain/commercial antigen could be specified, or a reference to a relevant methodology (English-written, pls.) could be provided.
P.2, l.196-7: “Samples were considered seropositive when IgG or IgM antibody titers were ≥1:256 and/or ≥1:16” – an ambiguous definition that should be made quite clear to the reader. Does it mean that ‘positive’ patients had jointly IgG ≥1:256 and IgM ≥1:16, or either IgG ≥1:256 or IgM ≥1:16 ? Note, pls., that the referenced methodological article in Korean ([18]) is of little aid to most readers.. Also, substitute “positive” for “seropositive”, pls.
P.3, l.225: Data was analysed..
P.3, l.228: delete the article before “multivariate”, pls.
P.3, l.230: ..set to..
P.3, l.234-5: in serology, the “cut-off ” is a PRE-SET diagnostic dilution discriminating between ‘positive’ and ‘negative’ sera. It is unique and constant for a given survey so the authors’ statement concerning its range/variation is nonsensical as I already noted. Perhaps, the authors wish to say something like this: “B.burgdorferi (s.lat.?)-specific IgG endpoint titres varied between <1:16 and 1:512, and in 21 samples (3.2%) reached or went beyond 1:256”, or so…
P.3, l.236: ditto
P.3, l.237: “..twenty-five samples (3.8%) had IgM titers of 1:16, and both had IgG titers of ≥1:256..” – confusing - “both” or “all” ?
P.3, l.244: .. the median duration of working in forest.. (?)
P.4, l.274: “seropositive to B.burgdorferi” rather than “seropositive for Lyme disease infection”
P.4, l.276: ..the level of education..
P.7, l.323: ..associated with a higher prevalence of..
P.7, l.325-6: ..taking a shower after outdoor work didn’t significantly reduce the risk of .. (?)
P.8, l.370-3: suggestion: (1) put a full stop after “on South Korea”, and (2) continue with “In the present analysis, we have identified risk factors associated with Lyme disease infection among NPOFWs, and completed the preliminary information in a poster presentation that..” (or simply: “..completed a preliminarily published information on this study [19]“).
P.8, l.373: ..from a part of this study..
P.8, l.373: An overall seroprevalence..
P.8, l.374-7: “The seroprevalence … using the same criteria” – reword this sentence to be smooth, pls.
P.8, l.384: ..that the amount of tick bites and the use of personal protective measures .. (?)
P.8, l.386: .. in that Belgian study.. ; .. paper reporting that..
P.8, l.389: ..was not associated with..
P.8, l.392: ..seroprevalence of..
P.8, l.396: ..share the environment with their.. ; .. infection in dogs..
P,8, l.397: delete the comma after “countries”, pls.
P.8, l.400: ..we found that..
P.9, l.462-3: .. though this association didn’t reach statistical significance in the multivariate analysis..
P.9, l.467: delete “that”, pls.
P.9, l.474: “least active” rather than “most inactive”
P.9, l.478: .. this is one of the most important studies conducted.. (indeed!?)
P.9, l.479: ..with a large sample.. ; ..the/our findings..
P.9, l.482: An overall seroprevalence..
P.9, l.500: ..contributions in conducting..
Table 1: delete the footnote “Ig, immunoglobulin”, it’s redundant (cf. p.2, l.195)
Author Response
Response to Reviewer 3 Comments
Point 1: I acknowledge that the most critical issues have been addressed. There remain, however, some minor ones that still need corrections.
Response 1: Thank you for reviewing this article and giving us careful comments. We have revised the manuscript based on the given comments. All changes have been indicated with red colored writings in the manuscript.
Pont 2: P.1. l. 47 vs. l.57 (and throughout the text): a small terminological inconsistency – “forestry-“ or “forest workers” ? Both terms are used indiscriminately in the text. Are those employee just forest craftsmen/lumberjacks or are they responsible for forest management/conservation in a broader sense? Perhaps a neutral term, e.g. “forest and conservation workers”, would fit…
Response 2: We unified the words into forestry workers throughout the text. Their jobs were explained in 2.2. Study settings and population (Line 62-73).
Pont 3: P.1., l.60: tested using immunofluorescence assay..
Response 3: We modified the sentence according to your comment (Line 13).
Pont 4: P.1, l.72: Lyme disease is a zoonotic tick-borne disease..
Response 4: We modified the sentence according to your comment (Line 25).
Pont 5: P.1, l.72-3: caused by a wide variety of the spirochaete..
Response 5: We modified the sentence according to your comment (Line 25).
Pont 6: P.1, l.87: comma after “acaricides”
Response 6: We modified the sentence according to your comment (Line 40).
Pont 7: P.2, l.163: obtaining an approval..
Response 7: We modified the sentence according to your comment (Line 57).
Pont 8: P.2, l.165: consents were obtained..
Response 8: We modified the sentence according to your comment (Line 59).
Pont 9: P.2, l.188: the enrolled workers were asked..
Response 9: We modified the sentence according to your comment (Line 82).
Pont 10: P.2, l.189: verified by the researchers..;
Response 10: We modified the sentence according to your comment (Line 83).
Pont 11: P.2, l.192: delete “also”, pls
Response 11: We modified the sentence according to your comment (Line 86).
Pont 12: P.2, l.196: “B. burgdorferi s.s.” (??) – an inhouse strain/commercial antigen could be specified, or a reference to a relevant methodology (English-written, pls.) could be provided.
Response 12: We provided the English-written reference about the methodology (Line 90, Line 265).
Reference: Park, S.H.; Hwang, K.J.; Chu, H.; Park, M.Y. Serologic detection of Lyme borreliosis agents in patients from Korea, 2005–2009. Osong Public Health Res. Perspect. 2011, 2, 29–33.
Pont 13: P.2, l.196-7: “Samples were considered seropositive when IgG or IgM antibody titers were ≥1:256 and/or ≥1:16” – an ambiguous definition that should be made quite clear to the reader. Does it mean that ‘positive’ patients had jointly IgG ≥1:256 and IgM ≥1:16, or either IgG ≥1:256 or IgM ≥1:16 ? Note, pls., that the referenced methodological article in Korean ([18]) is of little aid to most readers.. Also, substitute “positive” for “seropositive”, pls.
Response 13: Samples were considered seropositive when IgG ≥1:256 or IgM ≥1:16 (Line 91, Line 160). We presented the English-written reference (Line 92, Line 265) and substituted positive for seropositive.
Reference: Park, S.H.; Hwang, K.J.; Chu, H.; Park, M.Y. Serologic detection of Lyme borreliosis agents in patients from Korea, 2005–2009. Osong Public Health Res. Perspect. 2011, 2, 29–33.
Pont 14: P.3, l.225: Data was analysed..
Response 14: We modified the sentence according to your comment (Line 94).
Pont 15: P.3, l.228: delete the article before “multivariate”, pls.
Response 15: We modified the sentence according to your comment (Line 97).
Pont 16: P.3, l.230: set to..
Response 16: We modified the sentence according to your comment (Line 99).
Pont 17: P.3, l.234-5: in serology, the “cut-off” is a PRE-SET diagnostic dilution discriminating between ‘positive’ and ‘negative’ sera. It is unique and constant for a given survey so the authors’ statement concerning its range/variation is nonsensical as I already noted. Perhaps, the authors wish to say something like this: “B.burgdorferi (s.lat.?)-specific IgG endpoint titres varied between <1:16 and 1:512, and in 21 samples (3.2%) reached or went beyond 1:256”, or so…
Response 17: In IFA, the titer cutoff was not predetermined, and might be different among countries and researchers. For example, a seroepidemiologic study in China presented the seroprevalence based on IgG titer ≥1:128 (Line 162-165). And we clarified the sentence according to your comment (Line 108-110).
Pont 18: P.3, l.236: ditto
Response 18: We clarified the sentence according to your comment (Line 110-111).
Pont 19: P.3, l.237: “twenty-five samples (3.8%) had IgM titers of 1:16, and both had IgG titers of ≥1:256..” – confusing - “both” or “all” ?
Response 19: We mean that two samples had IgG titers of ≥1:256 and IgM titers of ≥1:16 (Line 111-112). We intended to explain the total number of seroprevalence (53; IgG: 21, IgM: 34, both IgG and IgM: 2).
Pont 20: P.3, l.244: the median duration of working in forest.. (?)
Response 20: We clarified the sentence according to your comment (Line 103).
Pont 21: P.4, l.274: “seropositive to B.burgdorferi” rather than “seropositive for Lyme disease infection”
Response 21: We modified the sentence according to your comment (Line 116).
Pont 22: P.4, l.276: the level of education..
Response 22: We modified the sentence according to your comment (Line 118).
Pont 23: P.7, l.323: associated with a higher prevalence of..
Response 23: We modified the sentence according to your comment (Line 146).
Pont 24: P.7, l.325-6: taking a shower after outdoor work didn’t significantly reduce the risk of .. (?)
Response 24: We modified the sentence according to your comment (Line 148-150).
Pont 25: P.8, l.370-3: suggestion: (1) put a full stop after “on South Korea”, and (2) continue with “In the present analysis, we have identified risk factors associated with Lyme disease infection among NPOFWs, and completed the preliminary information in a poster presentation that..” (or simply: “..completed a preliminarily published information on this study [19]“).
Response 25: We modified the sentence according to your comment (Line 156-158).
Pont 26: P.8, l.373: from a part of this study..
Response 26: We deleted the phrase (Line 156-158).
Pont 27: P.8, l.373: An overall seroprevalence..
Response 27: We modified the sentence according to your comment (Line 158).
Pont 28: P.8, l.374-7: “The seroprevalence … using the same criteria” – reword this sentence to be smooth, pls.
Response 28: We reword this sentence according to your comment (Line 160-162).
Pont 29: P.8, l.384: that the amount of tick bites and the use of personal protective measures ..(?)
Response 29: We modified the sentence according to your comment (Line 169-170).
Pont 30: P.8, l.386: in that Belgian study.. ; .. paper reporting that..
Response 30: We modified the sentence according to your comment (Line 171-172).
Pont 31: P.8, l.389: was not associated with..
Response 32: We modified the sentence according to your comment (Line 175).
Pont 32: P.8, l.392: seroprevalence of..
Response 32: We modified the sentence according to your comment (Line 178).
Pont 33: P.8, l.396: share the environment with their.. ; .. infection in dogs..
Response 33: We modified the sentence according to your comment (Line 182-183).
Pont 34: P,8, l.397: delete the comma after “countries”, pls.
Response 34: We modified the sentence according to your comment (Line 183).
Pont 35: P.8, l.400: we found that..
Response 35: We modified the sentence according to your comment (Line 187).
Pont 36: P.9, l.462-3: though this association didn’t reach statistical significance in the multivariate analysis..
Response 36: We modified the sentence according to your comment (Line 188-189).
Pont 37: P.9, l.467: delete “that”, pls.
Response 37: We modified the sentence according to your comment (Line 193).
Pont 38: P.9, l.474: “least active” rather than “most inactive”
Response 38: We modified the sentence according to your comment (Line 200).
Pont 39: P.9, l.478: this is one of the most important studies conducted.. (indeed!?)
Response 39: We modified “the most important” to “important” (line 204).
Pont 40: P.9, l.479: with a large sample.. ; ..the/our findings..
Response 40: We modified the sentence according to your comment (Line 205).
Pont 41: P.9, l.482: An overall seroprevalence..
Response 41: We modified the sentence according to your comment (Line 208).
Pont 42: P.9, l.500: contributions in conducting..
Response 42: We modified the sentence according to your comment (Line 226).
Pont 43: Table 1: delete the footnote “Ig, immunoglobulin”, it’s redundant (cf. p.2, l.195)
Response 43: We deleted the footnote in Table 1 according to your comment (Line 114).
This manuscript is a resubmission of an earlier submission. The following is a list of the peer review reports and author responses from that submission.
Round 1
Reviewer 1 Report
Acharya & Park present the results from a seroprevalence study for Borrelia burgdorferi, the causative agent of Lyme disease, antibodies in forestry workers in South Korea.
I have a few requests for authors to check on the wording and meaning of 2 sentences in the Introduction and to check data reported in the Results:
Statement in Introduction on lines 31-32 is incorrect. As authors will note upon review of reference 2 again, most people treated with antibiotics for Lyme disease do recover, whereas only a small percentage have persisting symptoms. Please rewrite the statement to reflect that.
On line 45, the rise in Lyme disease cases reported in South Korea may be better described as “a rise in cases”, rather than “growing incidence,” as written, since the numbers of cases annually are still very low, such that incidence, e.g., number of cases per 100,000 people, probably hasn’t changed much since reporting began in 2010.
Please check statement on line 107, “two samples (0.3%) had IgM titers of 1:16,” as Table 1 reports that 25 had IgM titers of 1:16.
I also would recommend that authors consider discussing the limitation that seropositivity is not necessarily equivalent to infection, but rather may simply be evidence of past exposure. It would be of interest to know if any of the seropositive forestry workers had evidence of previous Lyme disease symptoms. Also, it would be of interest to know how frequently forestry workers report tick exposures. Do they report finding ticks after working outdoors, and would those workers that engage in the activities found to be positively associated with seropositivity in the study, such as eating meals outdoors or raising dogs, report a higher frequency of tick exposure than workers that do not engage in such activities. Such points may be discussed in the Discussion.
Reviewer 2 Report
Lyme disease is a multi-systemic, infectious disease caused by bacteria. Human infection occurs after a tick bites and feeds in the skin. Numerous animal species, mainly rodents, are the reservoir of the bacteria.
Lyme disease is a serious disease that prevalence is increasing every year. That is why epidemiological supervision over Lyme disease and the implementation of preventive programs is very important. Forest workers, national parks workers and farmers are particularly at risk of falling ill.
The article is very well written, the methodology is adequate for the purpose of the research. The only remark concerns the Results section. For better clarity, please change the order. First, review Personal profiles of the participants. Then Serologic results, as well as logistic regression analysis results between demographic characteristics and Lyme disease (...)
Well done.
Thank you.
Reviewer 3 Report
Dilaram Acharya & Ji-Hyuk Park: Seroepidemiologic survey of Lyme disease among forestry workers in National Park Offices in South Korea
This is a concisely written manuscript that could have been a valuable contribution to the deficient knowledge of Lyme borreliosis prevalence in South Korea. However, it suffers from ethical issues. The authors incorrectly claim that the studies so far published in South Korea “did not investigate Lyme disease infection among forestry workers”, and that this is the first study aiming at it (p.1, l.49-51; p.8, l.158-9). In reality, there exists a prior communication entitled “Serological study on lyme disease in national park workers in South Korea”, published in 2017 in Int. J. Antimicrob. Agents, vol. 50, suppl.1, p. S152, which the authors don’t cite. That study, beyond question, deals with the same material (collected in December 2016), was supported by the same grant (Korea Centers for Disease Control and Prevention, #2016ER520100), but has been authored by a different team (K.M. Lee, Y.T. Noh, J.H. Park, Y.S. Lee, H.S. Lim & S.Y. Kim). If comparing the contents of both, it appears that the present study is an extension of the former one, from the authorship of which all contributors employed in laboratory serology (and perhaps in the material’s collection, as well) have been excluded (that is - in my opinion - those who make the greatest piece of work in this kind of studies). Even if assuming that the present authors have their consent (though this seems weird and no written consent in support of it is attached), according to the commonly accepted standards of the International Committee of Medical Journal Ethics, such co-workers should be (at least) acknowledged and their contribution specified (ICMJE’s “Recommendations for the Conduct, Reporting, Editing, and Publication of Scholarly Work in Medical Journals”, Section II.A.3). In addition, editors are advised to require that the corresponding author obtain written permission from them (ICMJE, ditto). In the light of this, I regret to recommend rejection of this manuscript in the present author arrangement.
Minor issues:
P.1, l.15 and throughout: using immunofluorescence assay
P-1, l.17 and throughout: an overall seroprevalence
P.2, l.83-4: when was the survey ever conducted? It doesn’t seem realistic that the “study team” (i.e. the two authors alone) managed to collect all the material within a month…
P.3, l. 105-6: “Titer cut-offs … ranged from <1:16 to 1:512” – nonsense statement (a cut-off is a threshold, not a range!)
P.3, l. 107: “Titer cut-105 offs … ranged from <1:16 to 1:32” - ditto
P.3, l.106-7: B.burgdorferi
P.3, l.116: “63.6” should read “63.7”; “36.4” should read “36.3”
Table 2: # of “High school or less” and # of “University or more” don’t sum to 655
Table 3: “Monitoring of natural resources” - # “Yes” + # “No” don’t sum to 655
“Repairing facilities” – ditto
“Supervision of illegal activities” – ditto
“Patrolling” – ditto
“Grass mowing” – ditto
“Cleaning” – ditto
Table 4: “Wearing a long-sleeved shirt” - # “Always” + # “Others” don’t sum to 655
“Wearing gloves” – ditto
“Wearing boots” – ditto
“Wearing a hat” – ditto
“Using insect repellents” – ditto
“Using a mat to rest” – ditto
“Eating meals in woodland” – ditto
“Taking a bath” – ditto
“Checking tick bites” – ditto
Table 5: “Recognition of tick bites” - # “Yes” + # “No” don’t sum to 655
Table 6, header: “..scrub typhus seroprevalence..” ??